# The Quantum Geometric Tensor in a Parameter-Dependent Curved Space

**DOI:** 10.3390/e24091236

**Published:** 2022-09-02

**Authors:** Joan A. Austrich-Olivares, Jose David Vergara

**Affiliations:** Departamento de Física de Altas Energías, Instituto de Ciencias Nucleares, Universidad Nacional Autónoma de México, Apartado Postal 70-543, Ciudad de México 04510, Mexico

**Keywords:** quantum phase transitions, quantum metric tensor, geometric phases, geometric quantum information

## Abstract

We introduce a quantum geometric tensor in a curved space with a parameter-dependent metric, which contains the quantum metric tensor as the symmetric part and the Berry curvature corresponding to the antisymmetric part. This parameter-dependent metric modifies the usual inner product, which induces modifications in the quantum metric tensor and Berry curvature by adding terms proportional to the derivatives with respect to the parameters of the determinant of the metric. The quantum metric tensor is obtained in two ways: By using the definition of the infinitesimal distance between two states in the parameter-dependent curved space and via the fidelity susceptibility approach. The usual Berry connection acquires an additional term with which the curved inner product converts the Berry connection into an object that transforms as a connection and density of weight one. Finally, we provide three examples in one dimension with a nontrivial metric: an anharmonic oscillator, a Morse-like potential, and a generalized anharmonic oscillator; and one in two dimensions: the coupled anharmonic oscillator in a curved space.

## 1. Introduction

Quantum information geometry is a recent approach to describing quantum information properties using geometry. If we consider pure states, one of the most used geometric measures is the quantum geometric tensor (QGT) introduced by Provost and Vallee in [1]. The real part of this tensor has been used to predict quantum phase transitions (QPTs) [2,3,4,5,6,7] and the imaginary part corresponds to the Berry curvature [8] which is an essential element to detect quantum interference [9,10]. Furthermore, the Berry curvature can be, in some cases, very useful to detect a QPT [11,12]. In the case of mixed states, the situation is more subtle because countless metric tensors satisfy the Fisher–Rao equivariance property [13,14,15,16]. In this case, introducing a metric tensor using a symmetric Jordan product could be the indicated procedure [17,18]. In the case of pure states, it is easy to check that the approach of Provost and Vallee is equivalent to the introduction of a Jordan product using the covariance matrix [19]. Because, for the moment, we are interested only in pure states, we will consider an extension of the work of Provost and Vallee [1]. To obtain this tensor, one needs to consider a family {ψ(λ)} of normalized vectors of some Hilbert space that depend smoothly on an *m*-dimensional real parameter $\lambda =\left(\lambda_{1},\ldots ,\lambda_{m}\right)\in R^{m}$ and on the computation of the infinitesimal distance between two states in the parameter space
(1)$$d\left(\psi (\lambda +\delta \lambda ),\psi (\lambda )\right)=∥\psi \left(\lambda +\delta \lambda \right)-\psi \left(\lambda \right)∥.$$
By requiring that this distance is invariant under the gauge transformation $\psi \to e^{-i\alpha (\lambda )}\psi$, one arrives at the QGT of the *n*-th state
(2)$$Q_{\rho \kappa }^{\left(n\right)}:=\left(\partial_{\rho }\psi_{n}|\partial_{\kappa }\psi_{n}\right)-\left(\partial_{\rho }\psi_{n}|\psi_{n}\right)\left(\psi_{n}|\partial_{\kappa }\psi_{n}\right),$$
where $\partial_{\rho }=\frac{\partial }{\partial \lambda_{\rho }}$ and the internal product is defined in the usual flat space
(3)$$\left(\phi (x,\lambda )|\psi (x,\lambda )\right)=\int_{Vol}d^{d}x\phi^{*}\left(x,\lambda \right)\psi \left(x,\lambda \right)$$
The (symmetric) real part of the QGT yields the quantum metric tensor (QMT) [1]
(4)Gρκ(n)=ReQρκ(n),
which is a Riemannian metric and provides the distance $\delta \ell^{2}=G_{\rho \kappa }^{\left(n\right)}\left(\lambda \right)\delta \lambda^{\rho }\delta \lambda^{\kappa }$ between the quantum states $\left|\psi_{n}\left(x,\lambda \right)\right)$ and $\left|\psi_{n}\left(x,\lambda +\delta \lambda \right)\right)$, corresponding to infinitesimally different parameters. The (antisymmetric) imaginary part of the QGT encodes the Berry curvature [8]
(5)Fρκ(n)=−2ImQρκ(n),
which, after being integrated over a surface subtended by a closed path in the parameter space, gives rise to Berry’s phase [8]. Moreover, (Equation 2) includes the Berry gauge connections defined by
(6)$$\beta_{\rho }\left(\lambda \right)=-i\left(\psi \left(\lambda \right)|\partial_{\rho }\psi \left(\lambda \right)\right).$$
From another perspective, one of the most interesting quantities in quantum information theory is the quantum fidelity [20], which corresponds to the modulus of the overlap between two pure states
(7)$$F\left(\psi^{\prime },\psi \right)=\left|\left(\psi^{\prime }|\psi \right)\right|$$
This fidelity is a valuable measure of the loss of information during the transportation of a quantum state over a long distance. Using the fidelity between two states that differ by an infinitesimal change in the parameters δλ, it is possible to recover the QMT (Equation 4) which is called the fidelity susceptibility [5].

An alternative definition to the QGT is to rewrite it in a perturbative form by inserting the identity operator $I=\sum_{m}\left|m\right)\left(m\right|$ in the first term of Equation (Equation 2) and using
(8)$$\left(m|\partial_{\rho }n\right)=\frac{\left(m|\partial_{\rho }\hat{H}|n\right)}{E_{n}-E_{m}}\;\mathrm{for}\;m\neq n,$$
which follows from the eigenvalue equation $\hat{H}\left|n\right)=E_{n}\left|n\right)$, then the QGT takes the form [4]
(9)$$Q_{\rho \kappa }^{\left(n\right)}=\sum_{m\neq n}\frac{\left(n\right|\partial_{\rho }\hat{H}\left|m)(m\right|\partial_{\kappa }\hat{H}\left|n\right)}{{\left(E_{m}-E_{n}\right)}^{2}}.$$
This expression shows that the singular points of the QGT can be associated with the QPT, which are characterized by the ground-state level crossing under the variation of some parameters of the system. However, this is only a heuristic argument that can be analyzed more carefully using the scaling properties of the QGT [11,12,21]. In general, it is clear from Equation (Equation 9) that the components of the QGT are singular at the points where the parameters take a value $\lambda^{*}\in M$ such that $E_{n}\left(\lambda^{*}\right)=E_{m}\left(\lambda^{*}\right)$.

Moreover, recent studies have shown that some materials present relevant changes in their electronic structure if they acquire a curvature [22]; even, in some cases, the curvature can induce new quantum phase transitions [23]. Consequently, it will be interesting to build a generalization of Provost and Vallee’s work by considering an internal product for a curved space where the metric may depend on some parameters. The main result of our work is this extension of the QGT in a curved space, and we show that the QMT acquires new relevant terms arising from the parameter-dependent metric. Furthermore, the Berry connection and curvature also have extra terms, and its properties under a general coordinate transformation change dramatically (Equation 23). The contents of this work are as follows: In Section 2, we propose the extension to curved space by using a geometric approach similar to Provost and Vallee’s. In Section 3, we explicitly build the Berry curvature and the quantum geometric tensor. In Section 4, we present two one-dimensional examples of the application of our procedure. In Section 5, we present a two-dimensional coupled anharmonic oscillator, with a curved metric. Section 6 contains an example with a Berry curvature different from zero. Finally, in Appendix A, we present an alternative deduction of the quantum metric tensor in curved space by computing the fidelity susceptibility.

## 2. Quantum Metric Tensor: Geometrical Approach

As we have mentioned in the introduction, we are interested in the case where the metric *does* depend on the parameters of the system, $\lambda \in R^{m}$. Then, the inner product must be replaced in a form that takes into account this dependence by introducing the square root of the determinant of the metric as the measure of the integral:(10)$$\left(\phi \left(\lambda \right),\psi \left(\lambda \right)\right)\to \langle \phi \left(\lambda \right)|\psi \left(\lambda \right)\rangle =\int_{Vol}d^{N}x\sqrt{g}\phi^{*}\left(\lambda \right)\psi \left(\lambda \right)$$
where $g=\det g_{ij}\left(x,\lambda \right)$ is the determinant of our N—dimensional configuration space metric. First of all, we have to note that the normalization condition 〈ψ|ψ〉=1 implies
(11)$$\partial_{\rho }\langle \psi |\psi \rangle =\langle \partial_{\rho }\psi |\psi \rangle +\langle \psi |\partial_{\rho }\psi \rangle -\frac{1}{2}\langle \sigma_{\rho }\rangle =0$$
where we define
(12)$$\sigma_{\rho }\equiv g_{\mu \nu }\partial_{\rho }g^{\mu \nu }$$
and ρ∈1,⋯,m. This quantity, which arises solely due to the curvature of the spatial metric, is responsible for the extra terms that appear in the generalization of the QGT. Because the measure of the inner product has been modified, and the metric may depend on the parameters of the system, we need to realize that the metric also must change by the variation of the parameter λ in a specific way. Therefore, the inner product should be read as
(13)$$\begin{array}{cc} \langle \phi (\lambda )|\psi (\lambda )\rangle & =\int_{Vol}d^{N}x{\left(g^{1/4}\left(\lambda \right)\phi \left(\lambda \right)\right)}^{*}\left(g^{1/4}\left(\lambda \right)\psi \left(\lambda \right)\right)\\ \; & \equiv \langle g^{1/4}\left(\lambda \right)\phi \left(\lambda \right)|g^{1/4}\left(\lambda \right)\psi \left(\lambda \right)\rangle \end{array}$$
where the $\sqrt{g}$ has been separated into two factors of $g^{1/4}$. This factorization of $\sqrt{g}$ is taken to consider the variation of the metric that corresponds to the state where the parameter has been shifted infinitesimally. It is easy to note that the volume element of this inner product remains invariant under coordinate transformations $x\to x^{\prime }=f\left(x\right)$, because the parameters λ do not depend on the coordinate *x*. Then, the metric transforms under the coordinate transformation as usual:(14)$$\begin{array}{c} g_{\mu \nu }\left(x^{\prime },\lambda \right)=\frac{\partial x^{\alpha }}{\partial x^{\mu }}\frac{\partial x^{\beta }}{\partial x^{\nu }}g_{\alpha \beta }\left(x,\lambda \right)\\ g_{\mu \nu }\left(x^{\prime },\lambda^{\prime }\right)=\frac{\partial x^{\alpha }}{\partial x^{\prime \mu }}\frac{\partial x^{\beta }}{\partial x^{\prime \nu }}g_{\alpha \beta }\left(x,\lambda^{\prime }\right). \end{array}$$
Thus, the determinant of the transformed metric will compensate the Jacobian, which arises from $d^{N}x$. Under the assumption of the adiabatic approximation, we can proceed to define a distance between a given state ψ(λ) and itself with shifted parameter $\psi (\lambda^{\prime })$ in its respective shifted parameter manifold in the following manner:(15)$$\begin{array}{cc} {∥\psi \left(\lambda +\delta \lambda \right)-\psi \left(\lambda \right)∥}^{2}= & 2-\langle g^{1/4}\left(\lambda +\delta \lambda \right)\psi \left(\lambda +\delta \lambda \right)|g^{1/4}\left(\lambda \right)\psi \left(\lambda \right)\rangle \\ \; & -\langle g^{1/4}\left(\lambda \right)\psi \left(\lambda \right)|g^{1/4}\left(\lambda +\delta \lambda \right)\psi \left(\lambda +\delta \lambda \right)\rangle \end{array}$$
Up to second order, this equation can be written in the following form:(16)$${∥g^{1/4}\left(\lambda +\delta \lambda \right)\psi \left(\lambda +\delta \lambda \right)-g^{1/4}\left(\lambda \right)\psi \left(\lambda \right)∥}^{2}=\gamma_{\rho \kappa }\delta \lambda^{\rho }\delta \lambda^{\kappa }$$
where
(17)$$\begin{array}{cc} \gamma_{\rho \kappa }\equiv & \;\;\frac{1}{2}\left(\langle g^{1/4}\partial_{\rho }\psi |g^{1/4}\partial_{\kappa }\psi \rangle +\langle g^{1/4}\partial_{\kappa }\psi |g^{1/4}\partial_{\rho }\psi \rangle \right)\\ \; & -\frac{1}{8}\left(\langle g^{1/4}\psi |\sigma_{\kappa }|g^{1/4}\partial_{\rho }\psi \rangle +\langle g^{1/4}\psi |\sigma_{\rho }|g^{1/4}\partial_{\kappa }\psi \rangle \right)\\ \; & -\frac{1}{8}\left(\langle g^{1/4}\partial_{\rho }\psi |\sigma_{\kappa }|g^{1/4}\psi \rangle +\langle g^{1/4}\partial_{\kappa }\psi |\sigma_{\rho }|g^{1/4}\psi \rangle \right)\\ \; & +\frac{1}{16}\langle \sigma_{\rho }\sigma_{\kappa }\rangle \end{array}$$
and $\langle \sigma_{\rho }\sigma_{\kappa }\rangle$ is the expectation value of $\sigma_{\rho }\sigma_{\kappa }$ with respect to our new definition of inner product: $\langle \sigma_{\rho }\sigma_{\kappa }\rangle \equiv \langle g^{1/4}\psi |\sigma_{\rho }\sigma_{\kappa }|g^{1/4}\psi \rangle$.

In an analogous way to [1], this tensor $\gamma_{\rho \kappa }$ is not invariant under the gauge transformation $\psi \to e^{i\alpha (\lambda )}\psi$. To incorporate this invariance in the metric, we introduce a modified Berry connection given by
(18)$$\beta_{\rho }=-i\langle g^{1/4}\psi |g^{1/4}\partial_{\rho }\psi \rangle +\frac{i}{4}\langle \sigma_{\rho }\rangle$$
which is real and, because $\langle \sigma_{\rho }\rangle$ is gauge invariant, it transforms as $\beta_{\rho }\to \beta_{\rho }+\partial_{\rho }\alpha$, under a gauge transformation.

Now, we can define a gauge-invariant symmetric tensor
(19)$$G_{\rho \kappa }=\gamma_{\rho \kappa }-\beta_{\rho }\beta_{\kappa }$$
which we also call the **quantum metric tensor** (QMT), given explicitly by
(20)$$\begin{array}{cc} G_{\rho \kappa }= & \frac{1}{2}\left(\langle g^{1/4}\partial_{\rho }\psi |g^{1/4}\partial_{\kappa }\psi \rangle +\langle g^{1/4}\partial_{\kappa }\psi |g^{1/4}\partial_{\rho }\psi \rangle \right)\\ \; & -\frac{1}{2}\left(\langle g^{1/4}\partial_{\rho }\psi |\psi \rangle \langle g^{1/4}\psi |g^{1/4}\partial_{\kappa }\psi \rangle +\langle g^{1/4}\partial_{\kappa }\psi |g^{1/4}\psi \rangle \langle g^{1/4}\psi |g^{1/4}\partial_{\rho }\psi \rangle \right)\\ \; & -\frac{1}{8}\left(\langle g^{1/4}\psi |\sigma_{\kappa }|g^{1/4}\partial_{\rho }\psi \rangle +\langle g^{1/4}\psi |\sigma_{\rho }|g^{1/4}\partial_{\kappa }\psi \rangle \right)\\ \; & -\frac{1}{8}\left(\langle g^{1/4}\partial_{\rho }\psi |\sigma_{\kappa }|g^{1/4}\psi \rangle +\langle g^{1/4}\partial_{\kappa }\psi |\sigma_{\rho }|g^{1/4}\psi \rangle \right)\\ \; & +\frac{1}{8}\left(\langle \sigma_{\rho }\rangle \langle g^{1/4}\psi |g^{1/4}\partial_{\kappa }\psi \rangle +\langle \sigma_{\kappa }\rangle \langle g^{1/4}\psi |g^{1/4}\partial_{\rho }\psi \rangle \right)\\ \; & +\frac{1}{8}\left(\langle \sigma_{\rho }\rangle \langle g^{1/4}\partial_{\kappa }\psi |g^{1/4}\psi \rangle +\langle \sigma_{\kappa }\rangle \langle g^{1/4}\partial_{\rho }\psi |g^{1/4}\psi \rangle \right)\\ \; & +\frac{1}{16}\langle \sigma_{\rho }\sigma_{\kappa }\rangle -\frac{1}{16}\langle \sigma_{\rho }\rangle \langle \sigma_{\kappa }\rangle \end{array}$$
In the expression above, we notice several additional terms, all related to the nontrivial metric introduced in the inner product, and it is reduced to the usual QMT in the flat-space limit.

## 3. Berry Curvature and Quantum Geometric Tensor

From the normalization condition (Equation 11), we define the Berry connection as
(21)$$\beta_{\rho }=-i\langle \psi |\partial_{\rho }\psi \rangle +\frac{i}{4}\langle \sigma_{\rho }\rangle .$$
First, we show that this quantity transforms as a connection under the transformation in the parameter space $\lambda \to \lambda^{\prime }$. We will consider that $\psi^{\prime }\left(\lambda^{\prime }\right)=\psi \left(\lambda \right)$. Then, $\sigma_{\rho }$ transforms as
(22)$$\langle \sigma_{\rho }^{\prime }\rangle =\left|\frac{\partial \lambda }{\partial \lambda^{\prime }}\right|\left(\frac{\partial \lambda^{\kappa }}{\partial \lambda^{\prime \rho }}\langle \sigma_{\kappa }\rangle +2\frac{\partial \lambda^{\alpha }}{\partial \lambda^{\prime \mu }}\frac{\partial^{2}\lambda^{\prime \mu }}{\partial \lambda^{\prime \rho }\partial \lambda^{\alpha }}\right)$$
Consequently, the Berry connection will transform as:(23)$$\beta_{\rho }^{\prime }=\left|\frac{\partial \lambda }{\partial \lambda^{\prime }}\right|\left(\frac{\partial \lambda^{\kappa }}{\partial \lambda^{\prime \rho }}\beta_{\kappa }+\frac{i}{2}\frac{\partial \lambda^{\alpha }}{\partial \lambda^{\prime \mu }}\frac{\partial^{2}\lambda^{\prime \mu }}{\partial \lambda^{\alpha }\partial \lambda^{\prime \rho }}\right)$$
Notice that $\beta_{\rho }$ transforms as a density connection of weight one. This last property appears because we integrate over the configuration space $x^{i}$ and not over the parameter space $\lambda^{\rho }$.

By definition, the Berry curvature is the exterior derivative of the Berry connection
(24)F=dβ,
and by a straightforward computation, we obtain the components of the Berry curvature:(25)$$\begin{array}{cc} F_{\rho \kappa } & =\partial_{\rho }\beta_{\kappa }-\partial_{\kappa }\beta_{\rho }\\ \; & =-i\left(\langle \partial_{\rho }\psi |\partial_{\kappa }\psi \rangle -\langle \partial_{\kappa }\psi |\partial_{\rho }\psi \rangle \right)+\frac{i}{4}\left(\langle \psi |\sigma_{\rho }|\partial_{\kappa }\psi \rangle -\langle \psi |\sigma_{\kappa }|\partial_{\rho }\psi \rangle \right)\\ \; & +\frac{i}{4}\left(\langle \partial_{\rho }\psi |\sigma_{\kappa }|\psi \rangle -\langle \partial_{\kappa }\psi |\sigma_{\rho }|\psi \rangle \right). \end{array}$$

With all of this in mind, we define the **quantum geometric tensor** (QGT), which combines the QMT and Berry curvature in one tensor:(26)$$G_{\rho \kappa }\equiv \langle \partial_{\rho }\left(g^{1/4}\psi \right)\left|P\right|\partial_{\kappa }\left(g^{1/4}\psi \right)\rangle$$
where P is a projection operator given by
(27)$$P=I-|g^{1/4}\psi \rangle \langle g^{1/4}\psi |.$$

Omitting the $g^{1/4}$ factors again, the QGT is explicitly given by
(28)$$\begin{array}{cc} G_{\rho \kappa }= & \langle \partial_{\rho }\psi |\partial_{\kappa }\psi \rangle -\langle \partial_{\rho }\psi |\psi \rangle \langle \psi |\partial_{\kappa }\psi \rangle -\frac{1}{4}\langle \psi |\sigma_{\rho }|\partial_{\kappa }\psi \rangle \\ \; & -\frac{1}{4}\langle \partial_{\rho }\psi |\sigma_{\kappa }|\psi \rangle +\frac{1}{4}\langle \sigma_{\rho }\rangle \langle \psi |\partial_{\kappa }\psi \rangle +\frac{1}{4}\langle \sigma_{\kappa }\rangle \langle \partial_{\rho }\psi |\psi \rangle \\ \; & +\frac{1}{16}\langle \sigma_{\rho }\sigma_{\kappa }\rangle -\frac{1}{16}\langle \sigma_{\rho }\rangle \langle \sigma_{\kappa }\rangle . \end{array}$$
The relevance of this tensor lies in the fact that it provides the fundamental structures underlying the parameter space: the symmetric part corresponds to the real part of (Equation 28),
(29)Re(Gρκ)=Gρκ
and the antisymmetric (imaginary) part yields the Berry curvature
(30)Im(Gρκ)=12Fρκ.
This is in correspondence to the QGT obtained in [1]. The QGT has the attractive property that it not only contains information about the frame bundle associated with the curvature of the parameter space but also contains information about the fiber bundle associated with the U(1) connection corresponding to the Berry phase. That both structures are contained in the QGT implies that this tensor contains all the information of the parameter space and its associated bundles. Another essential property of the QGT is that it allows the complete study of the symmetries of the quantum system. For example, it shows explicitly that the system is gauge invariant under phase transformations of the physical states.

## 4. Examples of the Quantum Metric Tensor in Curved Space

To consider some examples of our construction, we use a Lagrangian of form
(31)$$L=\frac{1}{2}g_{ij}\left(x,\lambda \right){\dot{x}}^{i}{\dot{x}}^{j}-V\left(x,\lambda \right),$$
that corresponds to a particle that is moving in curved space with a metric $g_{ij}\left(x,\lambda \right)$ that depends on some parameter λ, with Hamiltonian given by
(32)$$H=\frac{1}{2}g^{ij}p_{i}p_{j}+V.$$
Now, to build the Schrödinger equation in the coordinate representation, we introduce the Laplace–Beltrami operator [24],
(33)$$g^{ij}p_{i}p_{j}\to \nabla^{2}\psi =\frac{1}{\sqrt{g}}\frac{\partial }{\partial x^{j}}\left(\sqrt{g}g^{ij}\frac{\partial \psi }{\partial x^{i}}\right)$$
Moreover, we can decompose the metric into solder forms or “inverse tetrads” $e_{a}^{i}\left(x,\lambda \right)$ as follows
(34)$$g^{ij}=e_{a}^{i}e^{aj},\;\;\;\eta^{ab}=e^{aj}e_{j}^{b}$$
where the metric $\eta^{ab}$ is an N—dimensional flat-space metric, and $e_{j}^{b}$ is the tetrad or vierbein. In terms of solder forms, the Laplace–Beltrami operator is given by
(35)$$\nabla^{2}\psi =\frac{1}{e}\frac{\partial }{\partial x^{j}}\left(ee_{a}^{i}e^{aj}\frac{\partial \psi }{\partial x^{i}}\right)$$
where e is the determinant of the tetrad $e_{j}^{b}$. To start, we will consider systems in one-dimensional space.

### 4.1. Anharmonic Oscillator in One Dimensional Curved Space

Let us consider a one-dimensional anharmonic oscillator in curved space with a metric given by
(36)$$g=4\lambda x^{2}$$
In this case, the Lagrangian and Hamiltonian take the form
(37)$$L=2\lambda x^{2}{\dot{x}}^{2}-\frac{\omega^{2}}{2}\lambda x^{4},$$

(38)$$H=\frac{1}{8}\frac{p_{x}^{2}}{\lambda x^{2}}+\frac{\omega^{2}}{2}\lambda x^{4}.$$
We can obtain this system from the ordinary harmonic oscillator using a gauge transformation [25]. Using the Laplace–Beltrami operator (Equation 33), we can derive the time-independent Schrödinger equation:(39)$$\left(-\frac{\hbar^{2}}{8\lambda x^{2}}\frac{d^{2}}{dx^{2}}+\frac{\hbar^{2}}{8\lambda x^{3}}\frac{d}{dx}+\frac{\omega^{2}}{2}\lambda x^{4}\right)\psi_{n}\left(x\right)=E_{n}\psi_{n}\left(x\right)$$
with solutions given by
(40)$$\psi_{n}\left(x\right)=\frac{1}{\sqrt{2^{n}n!}}{\left(\frac{\omega }{\pi \hbar }\right)}^{1/4}e^{-\frac{\omega \lambda x^{4}}{2\hbar }}H_{n}\left(\sqrt{\frac{\omega \lambda }{\hbar }}x^{2}\right)$$
for n=0,1,2,… and $H_{n}\left(x\right)$ are the Hermite polynomials.

Moreover, the energy eigenvalues are the same as for the harmonic oscillator:(41)$$E_{n}=\hbar \omega \left(n+\frac{1}{2}\right)\;n=0,1,2,\ldots$$

Now, to compute the QGT, we note that ω and λ are the parameters of the system, and because there is no imaginary term in $\psi_{n}\left(x\right)$, we have that the *Berry curvature* is zero; thus, the quantum geometric tensor is the same as the quantum metric tensor. Now, we compute $\sigma_{\rho }$ as defined in (Equation 12) for ρ∈{λ,ω}
(42)$$\sigma_{\lambda }=-\frac{1}{\lambda }$$
(43)$$\sigma_{\omega }=0$$

Thus, we are able to compute the QMT for the *n*-excited state:(44)$$G\left[n\right]=\left(n^{2}+n+1\right)\left(\begin{array}{ccc} \frac{1}{8\lambda^{2}} & \; & \frac{1}{8\lambda \omega }\\ \frac{1}{8\lambda \omega } & \; & \frac{1}{8\omega^{2}} \end{array}\right)$$
where we denote with square brackets [n] the dependence on the quantum number *n*. As we have mentioned, the QGT and QMT are the same, and they are degenerated because the parameters (ω,λ) are not independent.

### 4.2. Harmonic Oscillator with a Morse Type Potential

To introduce one example with a $\sigma_{\rho }$ that is coordinate dependent, we choose another gauge-related potential to the harmonic oscillator. We will consider a potential that corresponds to the short-range repulsion term of the Morse potential but in a one-dimensional curved space with metric given by
(45)$$g=\frac{\lambda^{2}}{4}e^{-\lambda x},$$
which depends on the parameter λ and the configuration variable *x*. In this case, the classical action and Hamiltonian are given by
(46)$$S=\int dt\frac{1}{2}\left(\frac{\lambda^{2}}{4}e^{-\lambda x}{\dot{x}}^{2}-\omega^{2}e^{-\lambda x}\right)$$
(47)$$H=\frac{2}{\lambda^{2}}e^{\lambda x}p_{x}^{2}+\frac{\omega^{2}}{2}e^{-\lambda x}$$
In order to show the differences between this system and the harmonic oscillator, we graph the phase space for this system in Figure 1. In this way, Figure 1a shows the phase diagram for different energies, Figure 1b for different λ, Figure 1c for different ω, and in Figure 1d, the interesting symmetry that appears when changing the value λ→−λ. Interestingly, the loop gets “bigger” and wider for increasing energy, while increasing ω is the other way around. For increasing λ, we can see that the loop gets wider but approaches 0 in the *x* axis. Moreover, it presents the interesting fact that the loop gets inverted symmetrically by changing the sign of the value of λ. We observe also that the system has a singularity at x→∞ for λ>0 and at x→−∞ for λ<0.

Then, for this system, the time-independent Schrödinger’s equation is obtained from (33), with the metric given in (45),
(48)$$\left[-\frac{2\hbar^{2}}{\lambda^{2}}e^{\lambda x}\left(\frac{\lambda }{2}\frac{d}{dx}+\frac{d^{2}}{dx^{2}}\right)+\frac{\omega^{2}}{2}e^{-\lambda x}\right]\psi_{n}\left(x\right)=E_{n}\psi_{n}\left(x\right).$$

In this case, we will focus only on the ground-state $\psi_{0}\left(x\right)$ which is given by
(49)$$\psi_{0}\left(x\right)=Ae^{-\frac{\omega }{2\hbar }e^{-\lambda x}}.$$
with energy eigenvalue of $E_{0}=\frac{\hbar \omega }{2}$.

To obtain the normalization constant *A*, we use the relation $\langle \psi_{0}|\psi_{0}\rangle =1$, where this *bracket* is the inner product of the curved space, so that
(50)$$\begin{array}{cc} \langle \psi_{0}|\psi_{0}\rangle & =A^{2}\int_{-\infty }^{\infty }dx\left(\frac{\lambda }{2}e^{-\frac{\lambda }{2}x}e^{-\frac{\omega }{\hbar }e^{-\lambda x}}\right). \end{array}$$
If we perform the change in variable $u=e^{-\frac{\lambda }{2}x}$, and noticing that $u\overset{x\to -\infty }{\to }0$ and $u\overset{x\to \infty }{\to }-\infty$, we arrive to
(51)$$\begin{array}{cc} \langle \psi_{0}|\psi_{0}\rangle & =A^{2}\int_{0}^{\infty }e^{-\frac{\omega }{\hbar }u^{2}}du \end{array}$$
which is the usual harmonic oscillator constrained to the positive real line $R^{+}$. Thus, the normalization constant *A* is
(52)$$A=\sqrt{2}{\left(\frac{\omega }{\pi \hbar }\right)}^{\frac{1}{4}}$$
We need to point out that the normalization constant differs from the usual harmonic oscillator by a factor of $\sqrt{2}$.

Before continuing, we have to mention that the Berry curvature of this system is again zero. Thus, the QGT and QMT are the same.

Using (45), we compute $\sigma_{\rho }=g\partial_{\rho }g^{-1}$.
(53)$$\begin{array}{cc} \; & \sigma_{\lambda }=x-\frac{2}{\lambda }\\ \; & \sigma_{\omega }=0. \end{array}$$

One big difference from the previous system is that $\sigma_{\lambda }$ does depend on the coordinate *x*, so we are going to need all the terms in Equation (Equation 20).

The components of the QMT for the ground state are given by:(54)$$\begin{array}{cc} G_{\lambda \lambda }= & \langle \partial_{\lambda }\psi |\partial_{\lambda }\psi \rangle -\langle \partial_{\lambda }\psi |\psi \rangle \langle \psi |\partial_{\lambda }\psi \rangle +\frac{1}{2}\langle \sigma_{\lambda }\rangle \langle \partial_{\lambda }\psi |\psi \rangle -\frac{1}{2}\langle \partial_{\lambda }\psi |\sigma_{\lambda }|\psi \rangle \\ \; & +\frac{1}{16}\langle \sigma_{\lambda }^{2}\rangle -\frac{1}{16}{\langle \sigma_{\lambda }\rangle }^{2} \end{array}$$
(55)$$G_{\lambda \omega }=\langle \partial_{\lambda }\psi |\partial_{\omega }\psi \rangle -\frac{1}{4}\langle \partial_{\omega }\psi |\sigma_{\lambda }|\psi \rangle$$
(56)$$G_{\omega \omega }=\langle \partial_{\omega }\psi |\partial_{\omega }\psi \rangle$$
For completion, we write the components for the QMT for the ground state explicitly:(57)$$\begin{array}{cc} G_{\lambda \lambda }= & \frac{1}{16\lambda^{2}}\left[4+2\left(\gamma -4\right)\gamma +\pi^{2}+2\ln{\left(4\right)}^{2}+4\left(\gamma -2\right)\ln\left(\frac{4\omega }{\hbar }\right)\right.\\ \; & \left.+2\ln\left(\frac{\omega }{\hbar }\right)\ln\left(\frac{16\omega }{\hbar }\right)\right] \end{array}$$
(58)Gλω=116λω2−2γ+2erfωℏ+lnℏ216ω2+1π2G2,33,0ωℏ|1,10,0,32−G2,33,0ωℏ|1,10,0,12
(59)$$G_{\omega \omega }=\frac{1}{8\omega^{2}}$$
where γ is the Euler constant, erf(z) the error function, and $G_{p,q}^{m,n}\left(z|\begin{array}{c} a_{1},\ldots ,a_{n},a_{n+1},\ldots ,a_{p}\\ b_{1},\ldots ,b_{m},b_{m+1},\ldots ,b_{q} \end{array}\right)$ the MeijerG function.

In Figure 2, we show the graphics for the components of the QMT, where we can note that the components of $G_{\lambda \lambda }$ and $G_{\omega \omega }$ are positive (graphics Figure 2a and Figure 2b, respectively), while the component $G_{\lambda \omega }$ Figure 2c presents a change in sign. In the contour plot Figure 2d, we plot $G_{\lambda \omega }$ for different values of λ, it is shown that the ω axis is cut in the same point given by $\omega_{0}∼1.03716$, and in Figure 2e, we plot $G_{\lambda \omega }$ for different values of ω, and we can appreciate that in the critical point $\omega =\omega_{0}$, the curves change sign too. This critical point $\omega_{0}∼1.03716$, most appreciated in Figure 2d, seems to have an impact on the behavior of the system; thus, it can be related to Figure 1c where it makes softer the so abrupt increase and decrease in momentum and accelerating back to infinity for $\omega <\omega_{0}$ where $G_{\lambda \omega }$ is negative.

In addition, the QMT presents the particularity of being non-singular, that is, it has a determinant different from zero, in contrast with the QMT of the harmonic oscillator. Furthermore, all the components of the QMT show a quantum phase transition for ω=0 or λ=0 where the system changes to a free particle. This transition is in some sense equivalent to the one presented in the phase space in the limit x→±∞, which is nevertheless here observed for any velocity, whereas in the phase space exists only for null velocity.

The 3D-Plot of Figure 3a shows that the determinant tends to zero to increasing values of λ or ω and is always positive. In Figure 3b, we leave ω constant, while in Figure 3c, we fix λ to a constant value.

## 5. QGT Coupled Anharmonic Oscillator

For this example, we will consider a coupled anharmonic oscillator in curved space with spatial metric:(60)$$g=\left(\begin{array}{cc} a^{2}x^{2} & 0\\ 0 & b^{2}y^{2} \end{array}\right)$$
which is a diagonal matrix dependent explicitly on the parameters *a* and *b*. Then, the Lagrangian is
(61)$$L=\frac{1}{2}a^{2}x^{2}{\dot{x}}^{2}+\frac{1}{2}b^{2}y^{2}{\dot{y}}^{2}-\frac{k_{1}}{2}\left(\frac{a^{2}}{4}x^{4}+\frac{b^{2}}{4}y^{4}\right)-\frac{k_{2}}{2}{\left(\frac{a}{2}x^{2}-\frac{b}{2}y^{2}\right)}^{2}$$
so that the Hamiltonian is
(62)$$H=\frac{p_{x}^{2}}{2a^{2}x^{2}}+\frac{p_{y}^{2}}{2b^{2}y^{2}}+\frac{k_{1}}{2}\left(\frac{a^{2}}{4}x^{4}+\frac{b^{2}}{4}y^{4}\right)+\frac{k_{2}}{2}{\left(\frac{a}{2}x^{2}-\frac{b}{2}y^{2}\right)}^{2}.$$
Because we are interested in the QGT, we need to quantize this system. To do so, we will consider the Laplace–Beltrami operator (Equation 33):(63)$$\begin{array}{cc} \nabla^{2}\psi & =\frac{1}{\sqrt{abxy}}\left(\partial_{x}\left(\frac{by}{ax}\frac{\partial \psi }{\partial x}\right)+\partial_{y}\left(\frac{ax}{by}\frac{\partial \psi }{\partial_{y}}\right)\right)\\ \; & =\frac{1}{a^{2}x^{2}}\frac{\partial^{2}\psi }{\partial x^{2}}-\frac{1}{a^{2}x^{3}}\frac{\partial \psi }{\partial x}+\frac{1}{b^{2}y^{2}}\frac{\partial^{2}\psi }{\partial y^{2}}-\frac{1}{b^{2}y^{3}}\frac{\partial \psi }{\partial y} \end{array}$$

Thus, the time-independent Schrödinger equation is
(64)$$\begin{array}{cc} \hat{H}\Psi_{n}\left(x,y\right) & =-\frac{\hbar^{2}}{2a^{2}x^{2}}\frac{\partial^{2}\Psi_{n}\left(x,y\right)}{\partial x^{2}}+\frac{\hbar^{2}}{2a^{2}x^{3}}\frac{\partial \Psi_{n}\left(x,y\right)}{\partial x}-\frac{\hbar^{2}}{2b^{2}y^{2}}\frac{\partial^{2}\Psi_{n}\left(x,y\right)}{\partial y^{2}}+\frac{\hbar^{2}}{2b^{2}y^{3}}\frac{\partial \Psi_{n}\left(x,y\right)}{\partial y}\\ \; & \;+\frac{k_{1}}{2}\left(\frac{a^{2}x^{4}}{4}+\frac{b^{2}y^{4}}{4}\right)\Psi_{n}\left(x,y\right)+\frac{k_{2}}{2}{\left(\frac{ax^{2}}{2}-\frac{by^{2}}{2}\right)}^{2}\Psi_{n}\left(x,y\right)\\ \; & =E_{n}\Psi_{n}\left(x,y\right) \end{array}$$

The ground-state solution is given by
(65)$$\Psi_{0}\left(x,y\right)=A\exp\left[-\frac{\omega_{1}a^{2}x^{4}}{8}-\frac{\omega_{2}b^{2}y^{4}}{8}-\beta \frac{abx^{2}y^{2}}{4}\right]$$
with
$$\omega_{1}=\sqrt{k_{1}}=\omega_{2},\;\beta =\frac{1}{2}\left(\sqrt{k_{1}}-\sqrt{k_{1}+2k_{2}}\right)<0$$
and *A* is the normalization constant which will be obtained later.

The energy of this ground state is
(66)$$E_{0}=\frac{1}{2}\left(\omega_{+}+\omega_{-}\right)$$
where
$$\omega_{+}=\sqrt{k_{1}},\;\omega_{-}=\sqrt{k_{1}+2k_{2}}$$
are the frequencies of the normal modes. Then, we can write our ground-state solution as
(67)$$\Psi_{0}\left(U^{\left(+\right)},U^{\left(-\right)}\right)=A\exp\left[-\frac{1}{2}\left(\omega_{+}U^{\left(+\right)}2+\omega_{-}U^{\left(-\right)}2\right)\right]$$
where we have defined
(68)$$U_{\pm }=\frac{1}{\sqrt{2}}\left(\frac{ax^{2}}{2}\pm \frac{by^{2}}{2}\right).$$

Now, it is time to compute the normalization constant so we can compute the QGT. Note that the inner product in this case is given by
(69)$$\langle \psi |\phi \rangle =\int_{-\infty }^{\infty }\int_{-\infty }^{\infty }dxdy\sqrt{a^{2}b^{2}x^{2}y^{2}}\psi^{*}\left(x,y\right)\phi \left(x,y\right)$$
Then, for the ground state: (70)$$\begin{array}{cc} \langle \Psi_{0}|\Psi_{0}\rangle & =\int_{0}^{\infty }\int_{0}^{\infty }dxdy\;4ab\;xy\;A^{2}\exp\left[-\frac{\omega_{1}a^{2}x^{4}}{4}-\frac{\omega_{2}b^{2}y^{4}}{4}-\beta \frac{abx^{2}y^{2}}{2}\right]\\ \; & =\int_{0}^{\infty }\int_{-U^{\left(+\right)}}^{U^{\left(+\right)}}dU^{\left(+\right)}dU^{\left(-\right)}\;A^{2}\exp\left[-\left(\omega_{+}U^{\left(+\right)}2+\omega_{-}U^{\left(-\right)}2\right)\right]\\ \; & =\frac{4A^{2}\arctan\left(\sqrt{\frac{\omega_{-}}{\omega_{+}}}\right)}{\sqrt{\omega_{+}\omega_{-}}}\\ \; & =1. \end{array}$$

The change of the limits of integration and the factor of 4, arise from the definition $\sqrt{x^{2}}=\left|x\right|$. We need to note that the region of integration stopped to be the whole plane with the change in variables; instead, one only integrates on the region shown in Figure 4, which is the upper cone delimited by $U^{\left(+\right)}=\left|U^{\left(-\right)}\right|$.

Then, we have that the ground-state solution is given by
(71)$$\Psi_{0}\left(x,y\right)=\frac{{\left(k_{1}\left(k_{1}+2k_{2}\right)\right)}^{1/8}}{2\sqrt{\arctan{\left(1+\frac{2k_{2}}{k_{1}}\right)}^{1/4}}}\exp\left[-\frac{\sqrt{k_{1}}}{8}a^{2}x^{4}-\frac{\sqrt{k_{1}}}{8}b^{2}y^{4}-\frac{1}{2}\left(\sqrt{k_{1}}-\sqrt{k_{1}-2k_{2}}\right)\frac{abx^{2}y^{2}}{4}\right]$$
where we have written explicitly the parameters of the system: $\left\{k_{1},k_{2},a,b\right\}$.

Because the wave function does not have an imaginary part, the Berry curvature is zero; thus, the QGT is the same as the QMT. Because the algebraic expressions of the QMT do not give any clear information about the behavior of it, we show in Figure 5 the plots or, more specific, the projections of the components of the QMT. First thing to notice is that the plots of $G_{bk_{1}}$ and $G_{bk_{2}}$ are not to be found. This is because if one interchanges the parameters *a* and *b*, they are the same as $G_{ak_{1}}$ and $G_{ak_{2}}$, respectively, and this has a huge impact on the behavior of the QMT, making it singular. The second thing to note is the dependence of the components of the QMT on the parameters of the system, because $G_{k_{1}k_{1}}$, $G_{k_{2}k_{2}}$, and $G_{k_{1}k_{2}}$ only depend on the spring constant $k_{1}$ and the coupling constant $k_{2}$, while the rest of the components depend also on the parameter *a* and the component $G_{ab}$ depends on all four parameters $k_{1}$, $k_{2}$, *a*, and *b* with the peculiarity that we can interchange *a* and *b*. Thirdly, the component for $G_{ab}$ with a=−1 and b=1 is just a translation by $\frac{1}{2}$ of $G_{aa}$ with a=1. Finally, even though the plots for $G_{ak_{2}}$ with $k_{1}=1$ and $k_{2}=1$ look alike, their difference is not a trivial function. It is easy to observe that when $k_{1}$ and *a* go to zero, the components of the QMT go to infinity, but it is not the case when $k_{2}$ goes to zero. In fact, the term $G_{k_{1}k_{1}}∼\frac{1}{k_{1}^{2}}$ is recovering the expression for the usual anharmonic oscillator, see Section 4.1. This is not surprising because $k_{2}=0$ means that the system is decoupled. Now, the terms $G_{k_{1}k_{2}}$ and $G_{k_{2}k_{2}}$ also take the form of $∼\frac{1}{k_{1}^{2}}$, $G_{aa}∼\frac{1}{a^{2}}$, $G_{k_{i}a}∼\frac{1}{ak_{1}}$ when i=1,2, and $G_{ab}=0$. These results are not so unexpected because they mean that the QMT keeps some information on the dimension of the space of parameters, which is a purely quantum effect.

As mentioned before, the determinant of the QMT is zero, which can be avoided by setting one of the parameters *a* or *b* to be constant. This enables us to plot the subdeterminant of the QMT in Figure 6 where we denoted as a suffix the parameter *b* set as a constant. One can see that it is positive definite, and it diverges when $k_{1}$, or *a* approaches zero. However, when $k_{2}\to 0$, then this subdeterminant takes the form $∼\frac{1}{a^{2}k_{1}^{4}}$. Therefore, we can see how the QMT, more specifically the subdeterminant $DetQMT_{b}$, detects two different quantum phase transitions: a→0 the system collapse into a one-dimensional modified harmonic oscillator, such as in Section 4.1, and the more interesting case $k_{1}\to 0$, where at first glance one could think it becomes the linear coupled harmonic oscillator [26,27], but in reality, the system becomes a coupled harmonic oscillator with only two parameters [28]

.

## 6. Generalized Anharmonic Oscillator in Curved Space

A well-known example of a system with Berry curvature is the generalized harmonic oscillator [9,10,29] where the Hamiltonian is given by
(72)$$H=\frac{1}{2}\left[cx^{2}+b\left(xp+px\right)+ap^{2}\right]$$
and by a straightforward calculation, we obtain the Lagrangian:(73)$$L=\frac{1}{2a}{\dot{x}}^{2}-\frac{\Omega }{2}x^{2}-\frac{b}{2a}\left(x\dot{x}+\dot{x}x\right)$$
with
(74)$$\Omega =c-\frac{b^{2}}{a}$$

In our case, we are going to consider the generalized anharmonic oscillator with a=1, so that the determinant of the QGT for the generalized harmonic oscillator is different from zero, with spatial metric
(75)$$g=4\lambda x^{2}$$
and Hamiltonian
(76)$$H=\frac{ap^{2}}{8\lambda x^{2}}+\frac{b}{4}\left(xp+px\right)+\frac{c\lambda x^{4}}{2}$$
so that the Lagrangian is
(77)$$L=\frac{2\lambda x^{2}{\dot{x}}^{2}}{a}-\frac{b\lambda }{a}\left(x^{3}\dot{x}+\dot{x}x^{3}\right)-\frac{\Omega }{2}\lambda x^{4}$$

In this case, the time-independent Schrödinger equation is given by
(78)$$\left(-\frac{\hbar^{2}}{8\lambda }\left(\frac{1}{x^{2}}\partial_{x}^{2}-\frac{1}{x^{3}}\partial_{x}\right)-\frac{i\hbar bx}{2}\partial_{x}-\frac{i\hbar b}{2}+\frac{c\lambda x^{4}}{2}\right)\psi_{n}\left(x\right)=E_{n}\psi \left(x\right)$$
where the time-independent solutions are
(79)$$\psi_{n}\left(x\right)={\left(\frac{\omega }{\pi \hbar }\right)}^{1/4}\frac{1}{\sqrt{2^{n}n!}}e^{-\frac{\omega \lambda x^{4}}{2\hbar }}H_{n}\left(\sqrt{\frac{\omega \lambda }{\hbar }}x^{2}\right)e^{-\frac{ib\lambda x^{4}}{2\hbar }}$$
and the energy eigenvalues are the same as the generalized harmonic oscillator:(80)$$E_{n}=\left(n+\frac{1}{2}\right)\hbar \omega$$
where $\omega =\sqrt{c-b^{2}}$.

In this case, the metric *g* and $\sigma_{\rho }$ are the same as the Equations (Equation 36), (Equation 42) and (Equation 43). Then, the QMT for the n-excited state is
(81)$$G\left[n\right]=\left(n^{2}+n+1\right)\left(\begin{array}{ccc} \frac{c}{8\omega^{2}\lambda^{2}} & 0 & \frac{1}{16\omega^{2}\lambda }\\ 0 & \frac{c}{8\omega^{4}} & -\frac{b}{16\omega^{4}}\\ \frac{1}{16\omega^{2}\lambda } & -\frac{b}{16\omega^{4}} & \frac{1}{32\omega^{4}} \end{array}\right)$$
which is degenerated. In this case, the *Berry curvature* is different from zero. In fact, using Equation (Equation 25), we have that for the n-excited state, it is given by
(82)$$F_{\rho \kappa }\left[n\right]=\frac{2n+1}{16\omega^{3}\lambda }\left(\begin{array}{ccc} 0 & 2c & -b\\ -2c & 0 & -\lambda \\ b & \lambda & 0 \end{array}\right).$$
This curvature reduces to the Berry curvature of [9] in the limit λ→∞.

## 7. Discussion

In this paper, we have shown an extension of the QGT for curved spaces in which the metric may depend on the parameters of the system. The derivation of the QMT was conducted in two different manners: one in a geometrical way, extending the work of Provost and Vallee [1], and the second one via the fidelity susceptibility approach, which is shown in Appendix A. To obtain the QGT, we had to define a new Berry connection (Equation 18). This connection presents an extra term solely dependent on the metric of the curved space. This new term and the modification of the inner product are responsible for ensuring that the Berry connection transforms not only as a connection but also as a density of weight one. With this modified Berry connection, we computed the Berry curvature. Finally, the QGT is given, and as expected, it contains the QMT (symmetric/real part) and the Berry curvature (antisymmetric/imaginary part). It would be exciting to find out if, using the QGT, it is possible to extract some global information beyond the Chern character associated with the Berry curvature and the information contained in the Pontrjagin characteristics classes [30]. To show the consequences of how a nontrivial metric dependent on the parameters of the system affects the QMT and the Berry curvature, we provided four examples: three in one dimension and one in two dimensions. One interesting aspect of the one-dimensional examples is that they are isospectral, i.e., they have the same energy as the harmonic oscillator. Thus, we conclude that the energy eigenvalues are not enough to detect the particular system we are working with nor the QPTs that there might be. Another interesting point is that the example with a Morse-like potential has some similarities with the Liouville Quantum Theory on the Riemann sphere [31], and it will be interesting to use our procedure to compute the QGT in this case. Moreover, the generalization of our results to a perturbative form of the QGT to the curved background could be helpful in detecting critical points in the shape of figures of interest [32], because the Laplace–Beltrami operator in higher dimensions gives the Schrödinger equation without potential in the curved background. Finally, we want to extend this work both for relativistic cases and for mixed states in order to detect QPTs, e.g., for quantum black holes, perhaps in a similar way to the quantum complexity approach of Susskind [33]. 

## Figures and Tables

**Figure 1 entropy-24-01236-f001:**
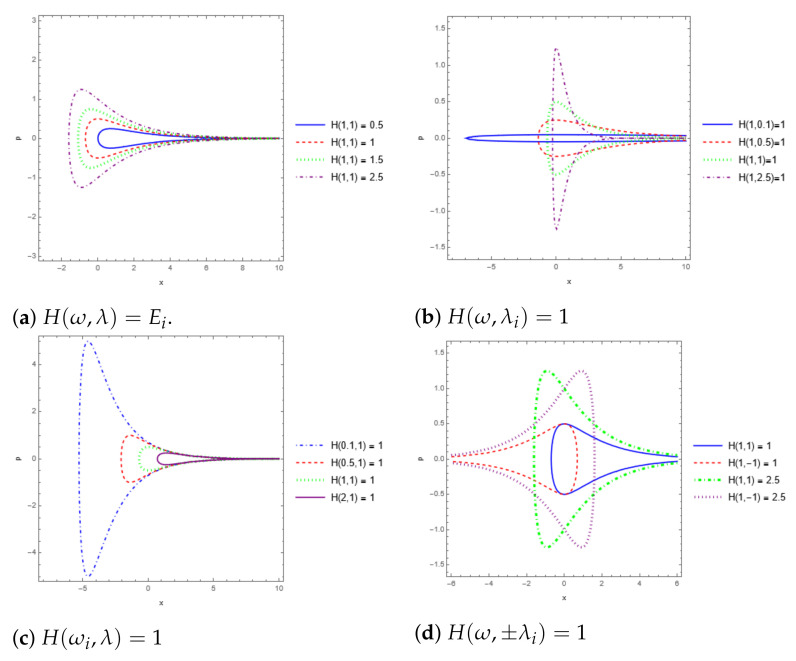
Phase diagrams for the Morse-like potential.

**Figure 2 entropy-24-01236-f002:**
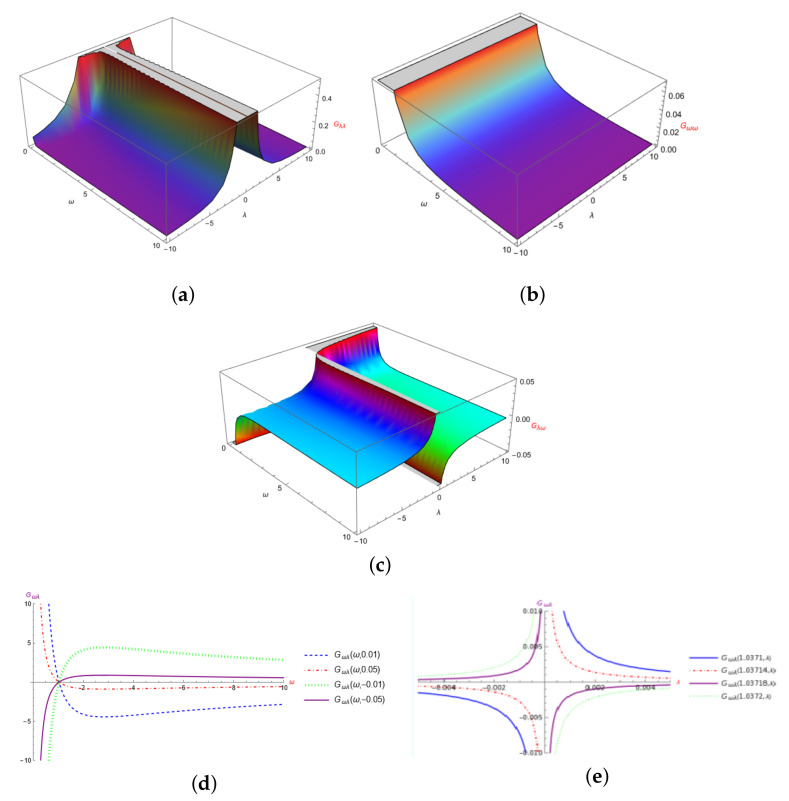
Components of the QMT for the Morse-like potential. (**a**) $G_{\lambda \lambda }$, (**b**) $G_{\omega \omega }$, (**c**) $G_{\lambda \omega }$, (**d**) Contour Plot of $G_{\omega \lambda }$, λ=(0.01,0.05,−0.01,−0.05), (**e**) Contour Plot of $G_{\omega \lambda }$, ω=(1.0371,1.03714,1.03718,1.0372).

**Figure 3 entropy-24-01236-f003:**
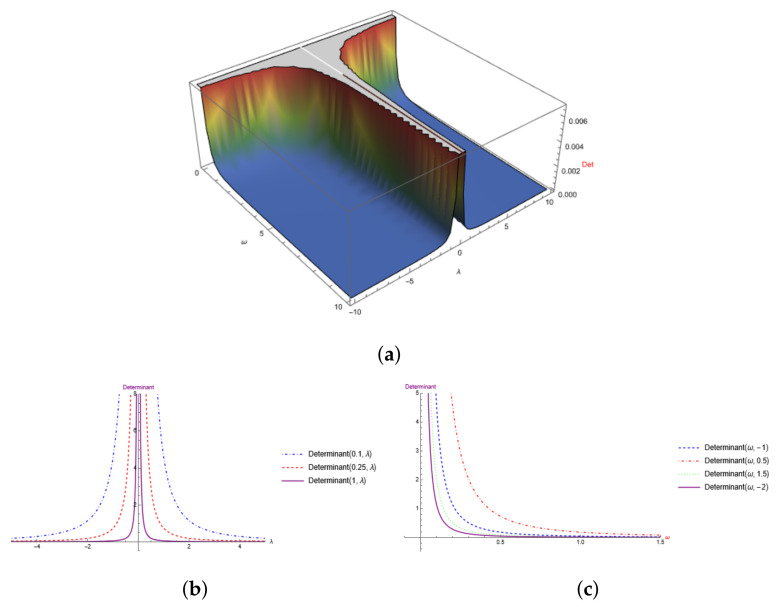
Determinant of the QMT for the Morse-like potential. (**a**) 3-D Plot of the determinant, (**b**) 2-D Projection of the determinant with ω constant. (**c**) 2-D Projection with λ fixed.

**Figure 4 entropy-24-01236-f004:**
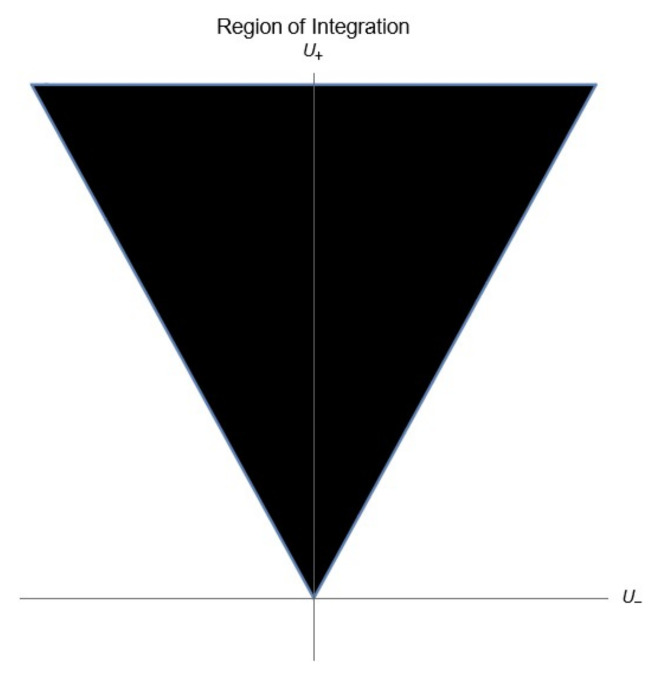
Region of Integration.

**Figure 5 entropy-24-01236-f005:**
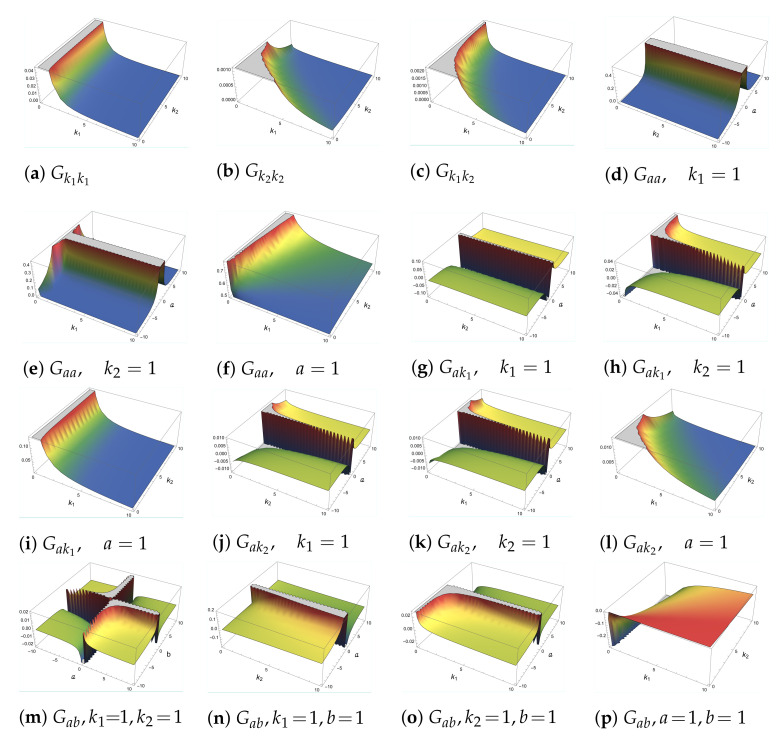
Components of the QMT for the coupled anharmonic oscillator in a curved space.

**Figure 6 entropy-24-01236-f006:**
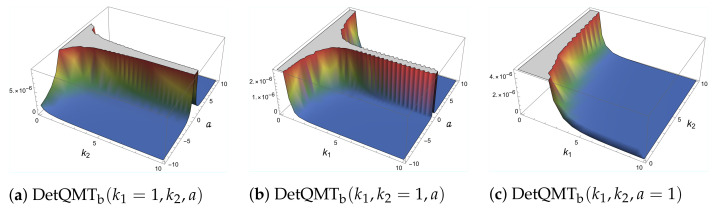
Subdeterminant of the QMT of the coupled anharmonic oscillator in a curved space.

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
