# Peer review of "The Quantum Geometric Tensor in a Parameter-Dependent Curved Space"

_entropy, 2022, doi:10.3390/e24091236_

Round 1

Reviewer 1 Report

The authors introduce quantum geometric tensor over the set of parameters that determine spatial metric of a system. They also present three examples in one and two in two dimensions. While the general idea seems interesting to me, there is one major problem upon which the whole paper depends, which was not considered by the authors. Therefore, I do not recommend the manuscript for publication in its current form. Below, I present my main doubt, upon which the whole work “hangs”.

The whole derivation of the new results crucially depends on the definition of the inner product between the state vectors for two different parameter values (for two different spatial metrics). While the explicit definition is not given in the manuscript, expressions (10) and (13) strongly suggests the following definition:

(*) < \psi(\lambda) | \psi(\lambda^\prime) >

= \int d^Nx [ g(\lambda) g(\lambda^\prime) ]^{1/4} \psi^*(\lambda) \psi(\lambda^\prime)

But the above definition might have two important issues:

1. What is the physical interpretation of the inner product between the state vectors corresponding to two different spatial metrics? For the state vectors having the same parameter value \lambda it has a standard interpretation, it is a probability amplitude to a system prepared in the state \psi to be found in the state \phi, given by equation (10) for the case of a general metric g(\lambda). But for different metrics, how do we interpret this? Note that the space is treated classical, so I am not sure one can even talk about quantum transitions between two different metrics.

2. More importantly, unlike the expression (10), which is invariant under the change of coordinates, x --> x^\prime = f(x), it is not clear that the above definition is. The authors must first prove the coordinate invariance of definition (*), as otherwise their results depend on the change of the (local) coordinate system(s), which is clearly not physical. I strongly suspect that the behaviour of the  fourth roots of the metrics determinants will "compensate“ the additional Jacobian coming from the transformation of d^Nx.

Author Response

We thank the referees for their detailed and valuable criticisms and suggestions, and positive comments, considering the research topic relevant. We have given proper attention to all the points mentioned by the referees, which helped improve the clarity and widen the perspective and implications of our results. We modify some sections and clarify some points.

{\bf Answering the specific points mentioned by the first referee:}

{\it The whole derivation of the new results crucially depends on the definition of the inner product between the state vectors for two different parameter values (for two different spatial metrics). While the explicit definition is not given in the manuscript, expressions (10) and (13) strongly suggests the following definition:

\begin{equation}\label{eq.def}

  \braket{\psi(\lambda) | \psi(\lambda^\prime) }

= \int d^Nx [ g(\lambda) g(\lambda^\prime) ]^{1/4} \psi^*(\lambda)

\psi(\lambda^\prime)

\end{equation}

Question1 :

What is the physical interpretation of the inner product between the state

vectors corresponding to two different spatial metrics? For the state vectors

having the same parameter value $\lambda$ it has a standard interpretation, it is a probability amplitude to a system prepared in the state $\psi$ to be found in the state $\phi$, given by equation (10) for the case of a general metric $g(\lambda)$. But for different metrics, how do we interpret this? Note that the space is treated classical, so I am not sure one can even talk about quantum transitions between two different metrics.}

Answer:

In this case, we are comparing the same state $\phi(x,\lambda)$ with itself, but with the parameters shifted $\phi(x,\lambda')$ where the parameters $\lambda$ do not depend on the coordinate $x$. Since we are taking a curved background with spatial metric dependent on the parameters, the shifted state will "live" in the shifted-parameter manifold; thus, the metric which is the same for the coordinate manifold $x$ will have a dependence on the shifted parameter $\lambda'$. We can think of this as we are in a sphere comparing a given state in a sphere with radius $r$ with itself in the sphere of radius $r' = r+\delta r$. 

\vskip2pc

{\it 

Question 2:

More importantly, unlike the expression (10), which is invariant under the

change of coordinates, $x \rightarrow x^\prime = f(x)$, it is not clear that the above definition is. The authors must first prove the coordinate invariance of definition \eqref{eq.def}, as otherwise, their results depend on the change of the (local) coordinate system(s), which is clearly not physical. I strongly suspect that the behavior of the fourth roots of the metrics determinants will "compensate“ the additional Jacobian coming from the transformation of $d^Nx$.}

Answer:

As mentioned before, the parameters $\lambda \in \mathbb{R}^m$ do not depend on the coordinate $x$.

Therefore the metric transforms under the coordinate transformation as usual:

\begin{equation}

\begin{split}

    g_{\mu\nu}(x',\lambda) = \frac{\partial x^\alpha}{\partial x^\mu}\frac{\partial x^\beta}{\partial x^\nu}g_{\alpha\beta}(x,\lambda) \\

     g_{\mu\nu}(x',\lambda') = \frac{\partial x^\alpha}{\partial x^\mu}\frac{\partial x^\beta}{\partial x^\nu}g_{\alpha\beta}(x,\lambda').

\end{split}

\end{equation} 

Thus, the determinant of the transformed metric will compensate the Jacobian, which arises from $d^Nx$. That is,

\begin{equation}

\begin{split}

    \int\limits_{Vol}d^{N}x' g^{1/4}(x',\lambda)g^{1/4}(x',\lambda') & = \int\limits_{Vol}d^{N}x \abs{\frac{\partial x'}{\partial x}}\left(\abs{\frac{\partial x^\mu}{\partial x'}}^{2} g(x,\lambda)\right)^{1/4}\left(\abs{\frac{\partial x}{\partial x'}}^{2}g(x,\lambda')\right)^{1/4} \\

    & =  \int\limits_{Vol}d^{N}x g^{1/4}(x,\lambda)g^{1/4}(x,\lambda')

\end{split}

\end{equation}

Reviewer 2 Report

Extending the study of physical quantities in the curved space may open up a new frontier particularly in nanoscale and quantum information science. I see this paper as paving the way for some interesting science in the gauge and geometric physics. On that merit, I recommend this paper to be published.

I have a minor query though. The Berry curvature has long been studied for the useful links it bears to many physical phenomena in condensed matter and quantum info science. It's definitely a good idea now to study the Berry curvature in the curved space.  On the other hand, the QMT can be derived from the distance between two states as mentioned in many parts of this paper.  The authors, however, also proposed at the start of the paper to study a QGT and identify in its real part the QMT and  in its imaginary part the Berry curvature.  But as both QMT and Berry curvature have in fact been studied in isolation, what is that extra merit of defining a QGT that only combines both towards the end in Eq.(25). In other words, how is it that a QGT constructed out of Berry curvature and QMT is a physically significant quantity in its own right.  

Author Response

We thank the referees for their detailed and valuable criticisms and suggestions, and positive comments, considering the research topic relevant. We have given proper attention to all the points mentioned by the referees, which helped improve the clarity and widen the perspective and implications of our results. We modify some sections and clarify some points.

{\bf Answering the specific points mentioned by the second referee:}

{\it Extending the study of physical quantities in the curved space may open up a new frontier particularly in nanoscale and quantum information science. I see this paper as paving the way for some interesting science in the gauge and geometric physics. On that merit, I recommend this paper to be published.}

{ \it Question 1

I have a minor query, though. The Berry curvature has long been studied for the useful links it bears to many physical phenomena in condensed matter and quantum info science. It's definitely a good idea now to study the Berry curvature in the curved space.  On the other hand, the QMT can be derived from the distance between two states as mentioned in many parts of this paper.  The authors, however, also proposed at the start of the paper to study a QGT and identify in its real part the QMT and in its imaginary part the Berry curvature.  But as both QMT and Berry curvature have in fact been studied in isolation, what is that extra merit of defining a QGT that only combines both towards the end in Eq.(25). In other words, how is it that a QGT constructed out of Berry curvature and QMT is a physically significant quantity in its own right.}

Answer: The QGT has the attractive property that it not only contains information about the frame bundle associated with the curvature of the parameter space but also contains information on the fiber bundle associated with the U(1) connection corresponding to the Berry phase, that both structures are contained in the QGT imply that this tensor contains all the information of the parameter space and its associated bundles. Another essential property of the QGT is that it allows the complete study of the symmetries of the quantum system. For example, it shows explicitly that the system is gauge invariant under phase transformations of the physical states.   

 It would be exciting to find out if, using the QGT, it is possible to extract some global information beyond the Chern character associated with the Berry curvature and the information contained in the Pontrjagin characteristics classes [Phys. Rep. 66(1980)213].
